# Illuminating protist diversity in pitcher plants and bromeliad tanks

**Robin S. Sleith**[1]*, **Laura A. Katz**[1,2]

**1** Smith College, Department of Biological Sciences, Northampton, Massachusetts, United States of America, **2** University of Massachusetts Amherst, Program in Organismic and Evolutionary Biology, Amherst, Massachusetts, United States of America

* rsleith@smith.edu

## Abstract

Many species of plants have evolved structures called phytotelmata that store water and trap detritus and prey. These structures house diverse communities of organisms, the inquiline microbiome, that aids breakdown of litter and prey. The invertebrate and bacterial food webs in these systems are well characterized, but less is known about microbial eukaryotic community dynamics. In this study we focus on microbes in the SAR clade (Stramenopila, Alveolata, Rhizaria) inhabiting phytotelmata. Using small subunit rDNA amplicon sequencing from repeated temporal and geographic samples of wild and cultivated plants across the Northeast U.S.A., we demonstrate that communities are variable within and between host plant type. Across habitats, communities from tropical bromeliads grown in a single room of a greenhouse were nearly as heterogeneous as wild pitcher plants spread across hundreds of kilometers. At the scale of pitcher plants in a single bog, analyses of samples from three time points suggest that seasonality is a major driver of protist community structure, with variable spring communities transitioning to more homogeneous communities that resemble the surrounding habitat. Our results indicate that protist communities in phytotelmata are variable, likely due to stochastic founder events and colonization/competition dynamics, leading to tremendous heterogeneity in inquiline microeukaryotic communities.

## Introduction

Microbial eukaryotes are critical and ubiquitous members of ecosystems and have profound impacts on natural and human systems [1–3]. These organisms are key components of food webs by serving in diverse roles such as autotrophs (i.e. diatoms, green algae), heterotrophs (i.e. most ciliates, amoebae) and mixotrophs (i.e. many dinoflagellates). Despite their tremendous biodiversity, only a limited number of molecular studies have focused on the biodiversity of plant-associated protist communities (e.g. [4–6]). Here we use a large-scale amplicon sequencing approach to target SAR (Stramenopila, Alveolata, Rhizaria) diversity in phytotelmata (i.e. water filled plant cavities). One of the largest clades of microbial eukaryotes, SAR (Stramenopila, Alveolata, Rhizaria) contains ~3/5ths of eukaryotic diversity (reviewed in [7]). While this clade includes well-studied model organisms such as *Plasmodium* and

**Data Availability Statement:** Raw reads associated with this study are available under NCBI BioProject ID: PRJNA682436.

**Funding:** This work is supported by an NSF grant DEB-1541511 to Laura A. Katz. The funders had no role in study design, data collection and analysis,

decision to publish, or preparation of the manuscript.

**Competing interests:** The authors have declared that no competing interests exist.

*Tetrahymena*, the diversity and ecology of much of this group remains to be discovered. Members of the SAR lineages are globally distributed, and large-scale marine sequencing projects have greatly expanded our understanding of the ecology of these organisms (e.g. [8]). Much less is known about SAR in freshwater habitats, which has been identified as an area in particular need of additional research [7].

Phytotelmata have evolved for a variety of functions including supplementing nutrients for plants growing in nutrient-poor environments (e.g. the pitcher plants and tank forming bromeliads that are the focus of this work). Two genera of pitcher plants, *Sarracenia* and *Nepenthes*, have evolved modified leaves that help them to survive in low nutrient bogs in North America and epiphytic habitats in Southeast Asia, respectively [9]. Species of *Nepenthes* (also known as tropical pitcher plants) secrete fluid and digestive enzymes that fill their traps whereas the traps of *Sarracenia* species fill with rainwater [10]. Epiphytic bromeliads are native to the New World, with a high diversity of species and life-history traits [11]. Tank-forming bromeliads are epiphytic bromeliad species adapted to store water and break down leaf litter and trapped organisms [4].

Pitcher plant and bromeliad phytotelmata, the focus of our study, have associated microbiomes that aid in the breakdown of captured prey and detritus [10, 12]. Initial investigations of the communities in phytotelmata of the purple pitcher plant (*Sarracenia purpurea*) demonstrate that these plants maintain a distinct food web where arthropods, protists, and bacteria interact to break down trapped animals/materials into host-available nutrients [10, 13]. A history of intense study spanning nearly three decades has established *Sarracenia purpurea* in particular as a model system for experimental ecology [12].

Much work has been done to investigate the bacterial and animal members of pitcher plant and bromeliad associated communities. The bacterial communities within phytotelmata of both *Sarracenia* and *Nepenthes* pitchers are heterogeneous in composition but distinct from surrounding aquatic or terrestrial communities [5]. Studies have shown that these bacterial communities have similar core functions (e.g., chitin degradation, protease and other extracellular enzyme production, amino acid metabolism) across populations; these functions are thought to contribute to a mutualistic relationship between plant and inquiline community [6, 14, 15]. The relationship between the bacterial community and the plant has furthermore been shown to be mediated by the larvae of the pitcher-plant mosquito (*Wyeomyia smithii*), which prey on bactivorous protists, thus increasing bacterial decomposition [13].

Studies of bacterial and animal communities in bromeliads detect communities distinct from surrounding habitats, with highly variable composition within and between host species [16–20]. In both the purple pitcher plant and bromeliad phytotelmata, mosquito larvae have been identified as a major controlling factor in food webs [21]. Gene sequencing and metatranscriptomic approaches have revealed genes associated with chitinases, nitrogenases, and methanogenesis, indicating that these habitats may be important for bacteria involved in global carbon and nitrogen cycling [17, 22].

In contrast to studies of bacteria and animals, few studies focus on the microbial eukaryotic communities in pitcher plants and bromeliads, though fungi have been the subject of several studies [23, 24]. Studies relying on morphological identification of protists have determined that their major role in these systems is as bactivores [25]. Stoichiometric models of these systems show that the relationship between the protists and the plant host can be parasitic [26], although there are conflicting accounts in the literature [27]. Previous studies of protists in phytotelmata typically relied on morphological identification (e.g. [28]), therefore focusing on a handful of identifiable protists and missing a vast portion of cryptic or unidentifiable diversity. The diversity of ciliates within tank forming bromeliads has been extensively studied, as these environments harbor high levels of rare and endemic species [4, 29, 30].

The few studies based on molecular techniques uncover interesting patterns that may further our understanding of the relationship between microbes (eukaryotic and prokaryotic) and the host plant. Satler et al. [31] found that the pattern of genetic diversity of eukaryotic inquiline community members was similar to the pattern of genetic diversity of the host plant (*Sarracenia alata*), indicating an evolutionary link (co-diversification) between plant and associated microbial community. Bittleston et al. [5] used an amplicon approach with "universal" eukaryotic primers to demonstrate that communities in two highly diverged pitcher plant genera were strikingly similar, and that pitcher plant communities represent a distinct subset of the community found in the background habitat. Simão et al. [4, 32] used a different primer set and found that ciliates were the most abundant protists in bromeliad phytotelmata. These approaches are advantageous in that a large portion of eukaryotic diversity may be captured, but problematic as members of certain eukaryotic lineages (e.g. Diptera) can swamp the signal of less dominant taxa in a sample and these 'universal' primers inevitably miss many lineages. Our taxon-focused approach here, an analysis of the SAR clade that represents a tremendous diversity of eukaryotes (Grattepanche et al. 2018), allows us to narrow our focus to comparisons of protist diversity between communities in space and time, and across ecological gradients. This focused approach is needed as few studies examine protist interactions with plants and even fewer draw observations from the natural environment [33].

In this study, we investigate the diversity of plant associated eukaryotic microbiomes to understand the community assemblage patterns in discrete freshwater ecosystems. We use an amplicon sequencing approach targeting the small ribosomal subunit, and we generate amplicons from RNA extractions to target active community members. In Study 1, we sample broadly across plant/phytotelmata type and landscape to ask how host plant type influences microbial community structure and diversity. This sampling spans natural and built environments, including cultivated tropical pitcher plants and bromeliads in the Lyman Conservatory at the Botanic Garden of Smith College and wild purple pitcher plants from bogs across the Northeast U.S.A. We hypothesize that samples from natural environments will host more species, and that communities will be structured by geography. In Study 2, we sample the same wild purple pitchers three times at a single site over the spring growing season to examine community stability and how seasonal changes lead to shifts in phytotelmata communities. We hypothesize that communities will diverge as the spring season progresses and colonization and dispersal increase.

## Materials and methods

### Sample collection—Study 1

For Study 1 we examined the influence of plant host on SAR diversity and community structure. Phytotelmata were sampled from three geographic locations across the Northeast, two in Massachusetts: Hawley Bog and the Lyman Conservatory, and one in Maine: Big Heath (see Table 1 for further locality information, permission was obtained at all sites prior to sampling). We sampled two dates in the Lyman Conservatory, repeating the same plants on the second date (3/21). We sampled Hawley Bog twice for a total of 17 samples (separate plants from Study 2) and Big Heath once for a total of 10 samples for Study 1 and sampled Hawley Bog three times for Study 2 (below). The contents of phytotelmata (up to 25mL) were collected using a sterile transfer pipette and either placed in a sterile tube or filtered immediately following the methods described below for Study 2. To sample the biofilm community present in the phytotelmata of pitcher plants and bromeliads we swabbed the interior wall of each with a sterile cotton applicator and swirled the applicator with the collected fluid to mix. Samples

**Table 1. Study and sampling information.**

| Study | Host | Species | Samples | Geography | Dates (all samples from 2019 Month/Day) |
|---|---|---|---|---|---|
| 1—Host | Bromeliad | *Aechmea* "Red Ribbon" | 3 | Lyman Conservatory, MA | 2/5; 3/21 |
| 1—Host | Bromeliad | *Nidularium innocentii* | 3 | Lyman Conservatory, MA | 2/5; 3/21 |
| 1—Host | Bromeliad | *Vriesea fosteriana* | 3 | Lyman Conservatory, MA | 2/5; 3/21 |
| 1—Host | Bromeliad | *Vriesea splendens* | 3 | Lyman Conservatory, MA | 2/5; 3/21 |
| 1—Host | Tropical Pitcher Plant | *Nepenthes maxima* | 4 | Lyman Conservatory, MA | 2/5; 3/21 |
| 1—Host | Tropical Pitcher Plant | *Nepenthes truncata* | 5 | Lyman Conservatory, MA | 2/5; 3/21 |
| 1—Host | Purple Pitcher Plant | *Sarracenia purpurea* | 17 | Hawley Bog, MA | 5/16; 6/27 |
| 1—Host | Purple Pitcher Plant | *Sarracenia purpurea* | 10 | Big Heath, Acadia National Park, ME | 5/21 |
| 1—Host + 2—Seasonal | Purple Pitcher Plant | *Sarracenia purpurea* | 27 | Hawley Bog, MA | 5/6 ("Day 1"); 5/29 ("Day 24"); 6/27 ("Day 53") |
| 1—Host + 2—Seasonal | Background Water | NA | 9 | Hawley Bog, MA | 5/6 ("Day 1"); 5/29 ("Day 24"); 6/27 ("Day 53") |

collected in tubes were returned to the lab and concentrated on 2μm filters using vacuum filtration. Each filter was placed in 500μl of RLT buffer (Qiagen) and stored at -80°C.

## Sample collection—Study 2

For Study 2 we examined how SAR diversity and community structure changes in *Sarracenia* pitchers over the course of the growing season at Hawley Bog, sampling nine pitchers three times from early May to late June. Pitchers at Hawley Bog emerge throughout the growing season (April—October), and bloom in late May to early June typically after their fifth growing season [12]. At this site, May and June receive an average of 11 cm and 9.5 cm of rain, respectively [34]. For repeated sampling, three rosettes of adult pitchers (likely including some from the previous growing season) were identified at the beginning, middle, and end of an established 26m transect extending from the forest edge to the center of the bog, closest to open water [35]. Three pitchers on each rosette were labelled using permanent marker and sampled on May 6th, May 29th, and June 27th 2019. The diameter of each pitcher opening was measured using calipers to understand the influence of pitcher size and morphology on the SAR community. For each sample, up to 25mL of fluid was removed using a sterile transfer pipette, volume was recorded, and fluid was concentrated on 2μm filters using a syringe and Swinnex Filter Holder. To sample the biofilm community present in the pitchers we swabbed the interior wall of each pitcher with a sterile cotton applicator and swirled the applicator with the collected fluid to mix. The filtrate was replaced in the pitcher to avoid pitchers desiccating prematurely, though we recognize that the amount of fluid in these pitchers varies tremendously throughout the growing season depending on rain levels, heat, cloudiness, and other weather factors. Each filter was placed in 500μl of RLT buffer (Qiagen) and stored at -80°C upon return from the field. To compare pitcher SAR communities to the background communities of the bog, 25mL of water below each rosette was collected and filtered using the methods described above.

## Sample processing—Studies 1 and 2

RNA was extracted using the Qiagen RNeasy kit. We removed DNA from the extracted RNA with the TURBO DNA-free™ Kit (Invitrogen, CA, USA), and generated single-strand cDNA

using the SuperScript® III First-Strand Synthesis System (Invitrogen) with random hexamer primers (Thermofisher, USA) following the methods of Sisson et al. [36].

We followed the methods of Sisson et al. [36] to generate amplicon libraries from the cDNA using SAR specific primers targeting the V3 region of the small ribosomal subunit (SSU-rRNA). Each PCR was conducted in triplicate then pooled to reduce PCR bias [37, 38]. The University of Rhode Island prepared sequencing libraries from amplicons and performed paired end (2x300bp) Illumina MiSeq High-Throughput Sequencing.

### Data analysis

Sequence reads were analyzed following the methods of Sisson et al. [36] with scripts available at https://github.com/jeandavidgrattepanche/Amplicon_MiSeq_pipeline. This pipeline quality filters and merges reads using PEAR [39], builds OTUs with SWARM v2 with the parameter d = 1 [40], and removes non-SAR 'outgroup' sequences using a phylogenetic approach in which OTUs are added to alignments based on full length reference sequences from GenBank (though the SAR primers target this group, a small proportion of non-SAR sequences are amplified due to the conserved nature of the SSU-rRNA). We subsampled each amplicon library to 5,000 reads and only included libraries with greater than 2,900 reads in subsequent analyses. We removed OTUs that had fewer than 100 reads in a single sample. The phylogenetic tree was created using RAxML [41] from a Mafft alignment [42] on the CIPRES Science Gateway [43]. We calculated dissimilarity matrices with UniFrac distances [44], both weighted (relative abundance) and unweighted (presence/absence), and performed principal coordinate analysis using R packages phyloseq and vegan [45–47]. Permutational multivariate analyses of variance (function adonis in vegan) were used to test for differences in community composition driven by host plant and sample date. Rarefaction curves were generated using the methods of Hausmann et al. [48]. The network analysis was conducted in R using RAM: R for Amplicon-Sequencing-Based Microbial-Ecology [49].

## Results

### Amplicon analysis of phytotelmata (Studies 1 and 2)

Based on analyses of 84 samples (54 *Sarracenia* pitcher plants, 9 *Nepenthes* pitcher plants, 12 bromeliads, 9 background water samples; see Table 1 for sample information), we found 463 OTUs (i.e. rDNA amplicons) represented by 420,116 reads that fall among SAR reference sequences using the phylogenetic approach described in Sisson et al. [36]. After removing samples with fewer than 2,900 reads and low abundance OTUs (i.e. fewer than 100 reads for a single sample, see methods), 135 OTUs represented by 391,044 reads in 81 samples remain for subsequent analyses (Fig 1, S2 File). Raw reads associated with this study are available under NCBI BioProject ID: PRJNA682436. The phylogenetic tree in Fig 1 shows that ciliates make up nearly half the 135 OTUs recovered in this study.

### Study 1: The influence of plant host on OTU richness and community structure

Rarefaction analysis of observed OTUs and Shannon's diversity index consistently rank samples from background bog water as more diverse than phytotelmata (Fig 2). We investigated whether plant type influences microbial community composition by comparing patterns across three distantly-related plant groups with convergent water trapping morphologies–bromeliads, *Nepenthes*, and *Sarracenia*. Although we detect the influence of host plant type on community structure (adonis: $R^2$ = 0.23, p = 0.001), we also observe tremendous heterogeneity

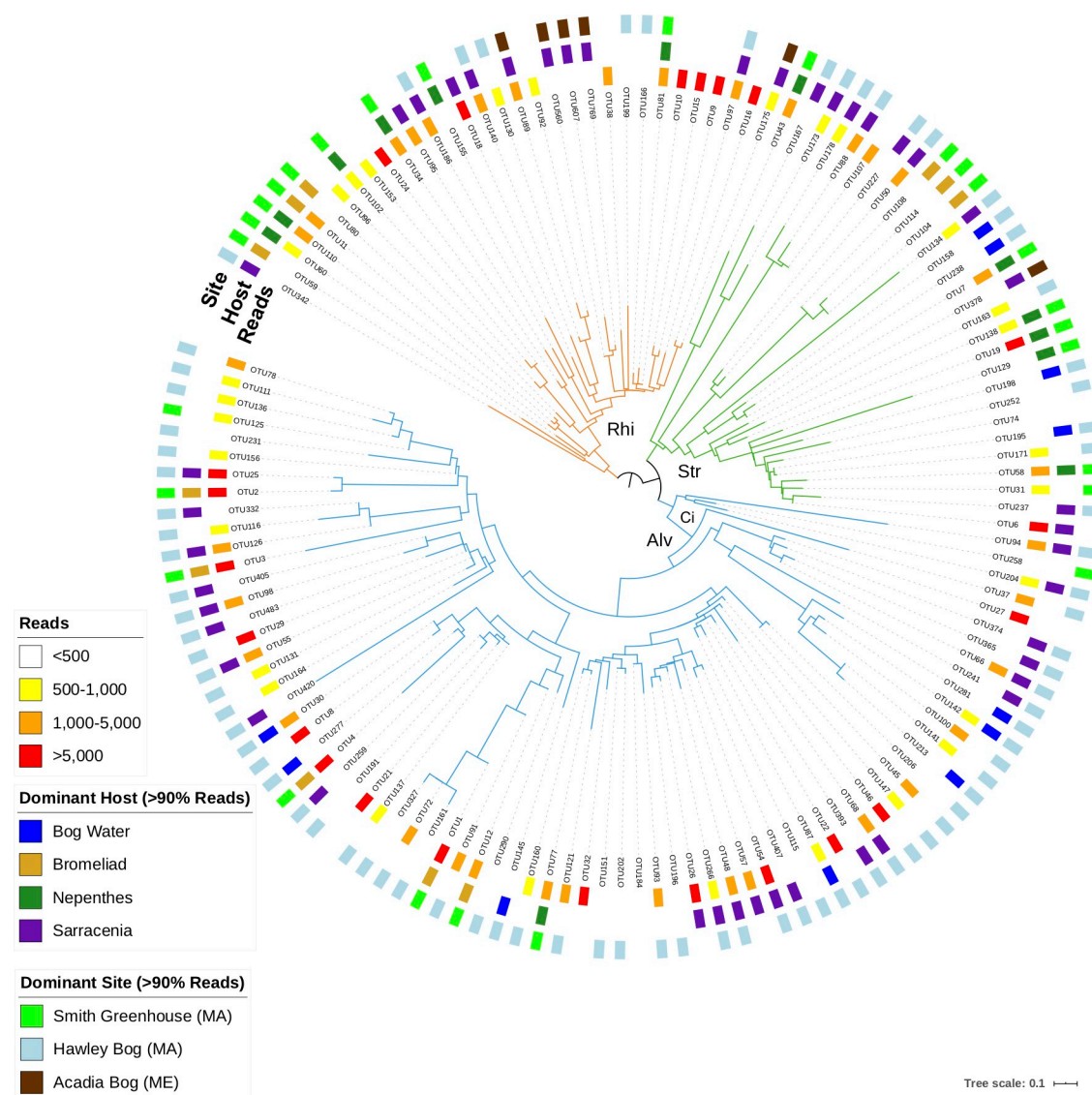

**Fig 1. The 135 focal OTUs from across plant hosts and geography illustrate generalist OTUs (i.e. not restricted by plant host or geography) and specialist OTUs (i.e. restricted by plant host or geography).** Branches are colored by major clades, Rhizaria (Rhi, orange) Stramenopila (Str, green) and Alveolata (Alv, blue, with Ci indicating ciliates). The number of reads is indicated in the inner ring, followed by host plant specificity (>90% of reads from a given host), followed by geographic specificity (>90% of reads from a given location).

among samples (Fig 2), with no clear geographic signal, counter to our hypothesis. Principal coordinate analysis based on weighted UNIFRAC distances shows overlap of communities in bromeliads and tropical pitcher plants growing in the Lyman Conservatory with those occurring in *Sarracenia purpurea* in two bogs across the Northeast U.S.A. (Fig 2). Communities from *Sarracenia purpurea* are most variable (i.e. occupy the most ordination space), followed by cultivated bromeliads and the pitcher plant *Nepenthes*. Although *Sarracenia* pitcher plants displayed the most heterogeneity, communities in bromeliad tanks located in a single room at the Lyman Conservatory occupy nearly as much ordination space (i.e. distance along PC1) as do communities occurring in *Sarracenia purpurea* in four bogs separated by hundreds of kilometers across the Northeast U.S.A. (Fig 2).

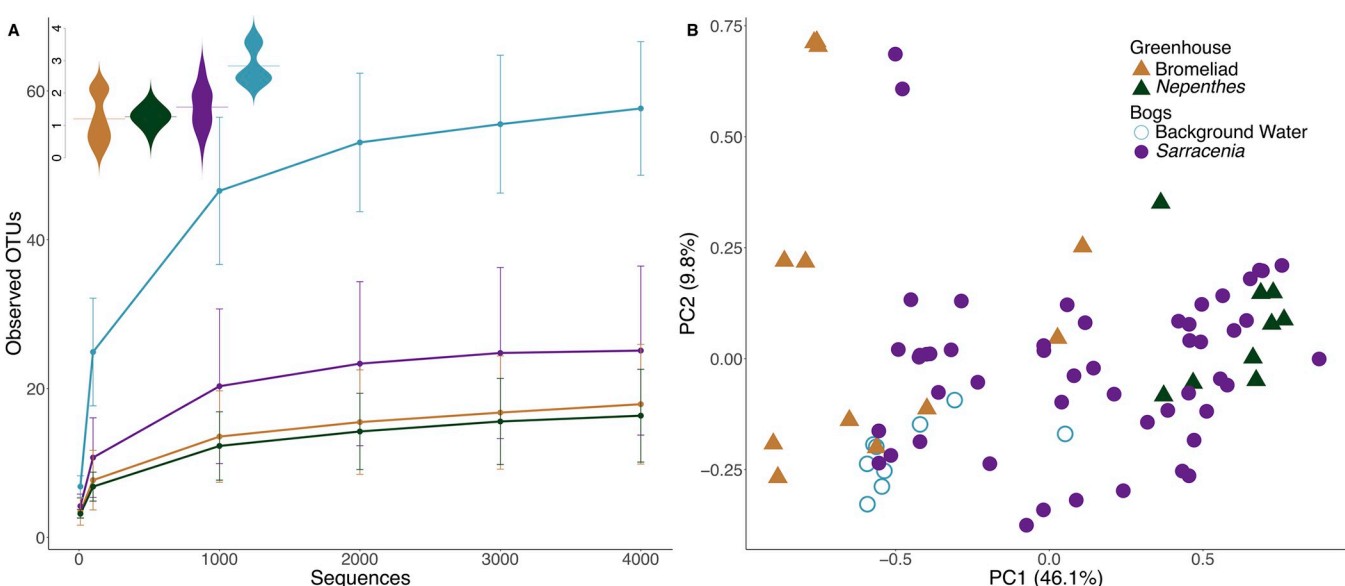

**Fig 2.** **A**. OTU richness (rarefaction plots, error bars are standard deviations) and Shannon diversity (inset beanplots) is highest in samples from the background community (light blue), followed by samples collected from *Sarracenia* (purple), bromeliads (brown), and *Nepenthes* (green). **B**. Principal Coordinate Analysis based on weighted UNIFRAC demonstrates differentiation of communities by host plant, with tremendous heterogeneity among communities in field-collections from *Sarracenia* (purple dots) and among greenhouse bromeliads (brown triangles; all from the same room in the Lyman Conservatory). Details on site and OTUs can be found in S1 and S2 Files.

## Study 2: The influence of season on OTU richness and community structure

We sampled nine pitchers three times at Hawley Bog over the course of the 2019 spring growing season (May 6th ("Day 1"); May 29th ("Day 24"); June 27th ("Day 53")), removing most but not all of the fluid each time. Analyses of the resulting amplicons show that the background samples consistently have more OTUs than the communities sampled in *Sarracenia* pitchers (S1 Fig). The highest richness estimated by rarefaction is in samples collected from the background water, with samples from the site closest to the forest edge consistently higher than the other sampling sites (i.e. rarefaction curve for site 1 background samples higher than all other curves in S1 Fig).

To determine whether pitcher volume or shape influenced OTU richness, we fit a linear model of observed OTU richness by pitcher diameter and volume (S2 Fig). We do not detect any pattern between the volume or the diameter of the top of pitchers and observed richness at Hawley Bog (p >0.7, S2 Fig). We find that the communities in *Sarracenia purpurea* at Hawley Bog are a subset of the bog water community as the network analysis shows connections between the OTUs occurring in pitchers and the background samples (Fig 3).

We again use principle coordinate analysis to determine how time and local habitat influence protist communities. We detect a temporal pattern whereby samples collected in early and late May vary widely across ordination space, while samples collected in late June (Day 53) cluster together on the right side of PC1 (Fig 4A, adonis: $R^2$ = 0.18, p = 0.003). The samples collected in late June also fall close to the background bog water samples, indicating that by late June the SAR community in the pitcher plants resembles the community of the background habitat. The OTU ordination in Fig 4B demonstrates that Alveolate taxa are the most abundant in samples from the background and late June, while many rhizarian taxa are abundant in the earlier months (as weighted Unifrac takes relative abundance into account in our ordination).

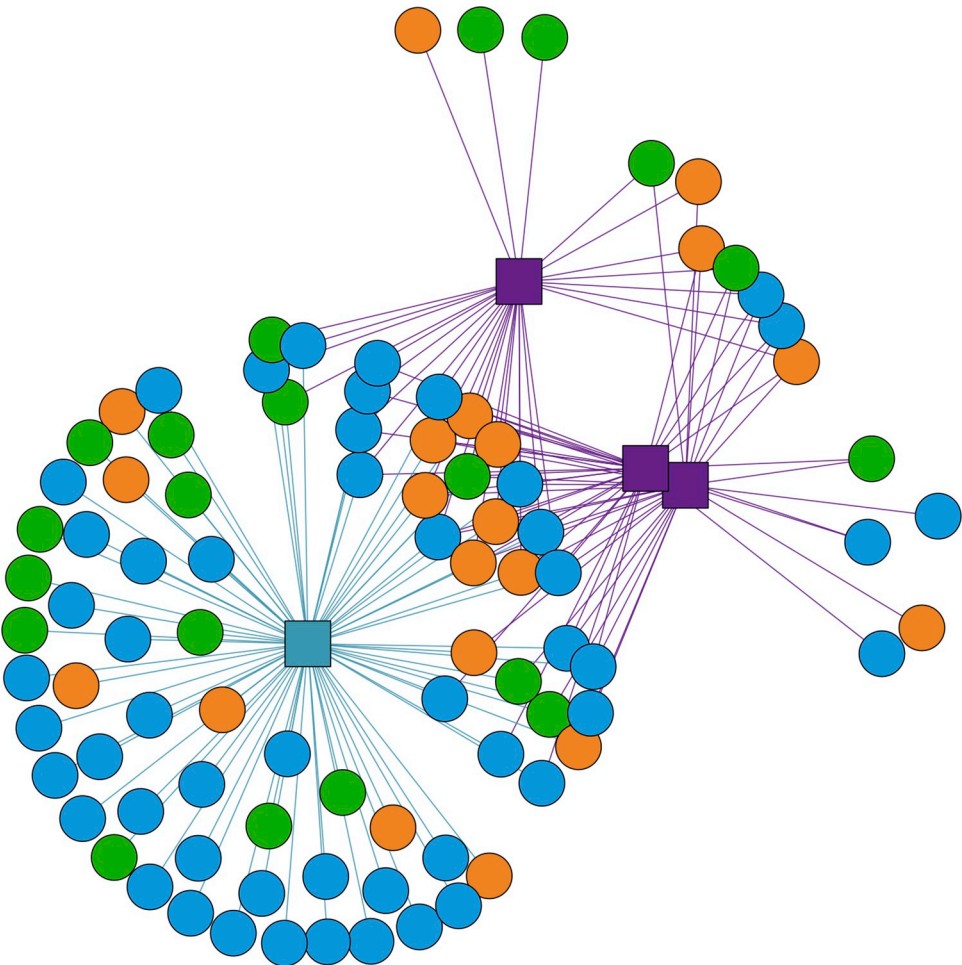

**Fig 3. Network analysis demonstrates that most OTUs in a rosette of 3 pitchers are shared with the bog water below the rosette.** Connection of OTUs (circles) shared (lines) between pitchers (purple squares) and bog water (blue square) for the rosette of pitchers at Hawley Bog closest to the forest edge, sampled on May 6th ("Day 1"). Taxonomy of OTUs is colored as Rhizaria (orange) Stramenopila (green) and Alveolata (blue).

We also detect temporal shifts in community structure at the highest taxonomic level (i.e. Stramenopila, Alveolata, Rhizaria), within *Sarracenia* pitchers. Over the three sampling times, OTUs identified as Stramenopila and Rhizaria decreased and OTUs identified as Alveolata increased across the spring season (Fig 5). The Alveolate OTUs with the largest increases across the growing season (OTU22, OTU25) have phylogenetic taxonomic assignment in the ciliate classes Colpodea and Oligohymenophorea, respectively (closest reference genera are *Bardeliella* and *Tetrahymena*, with full taxonomic information in S2 File, columns G and H). The Rhizarian OTU with the largest decrease (OTU10 in Figs 1 and 4) is in the Viridiraptoridae, a parasite of algae. The Stramenopile OTU with the largest decrease (OTU6 in Figs 1 and 4) is an autotrophic Chrysophyte. In the background community, OTU8, a ciliate most closely related to *Leptopharynx* was dominant.

## Discussion

The main insights of this study include: 1) inquiline SAR communities contain fewer OTUs than nearby aquatic communities, irrespective of plant host 2) plant host influences SAR

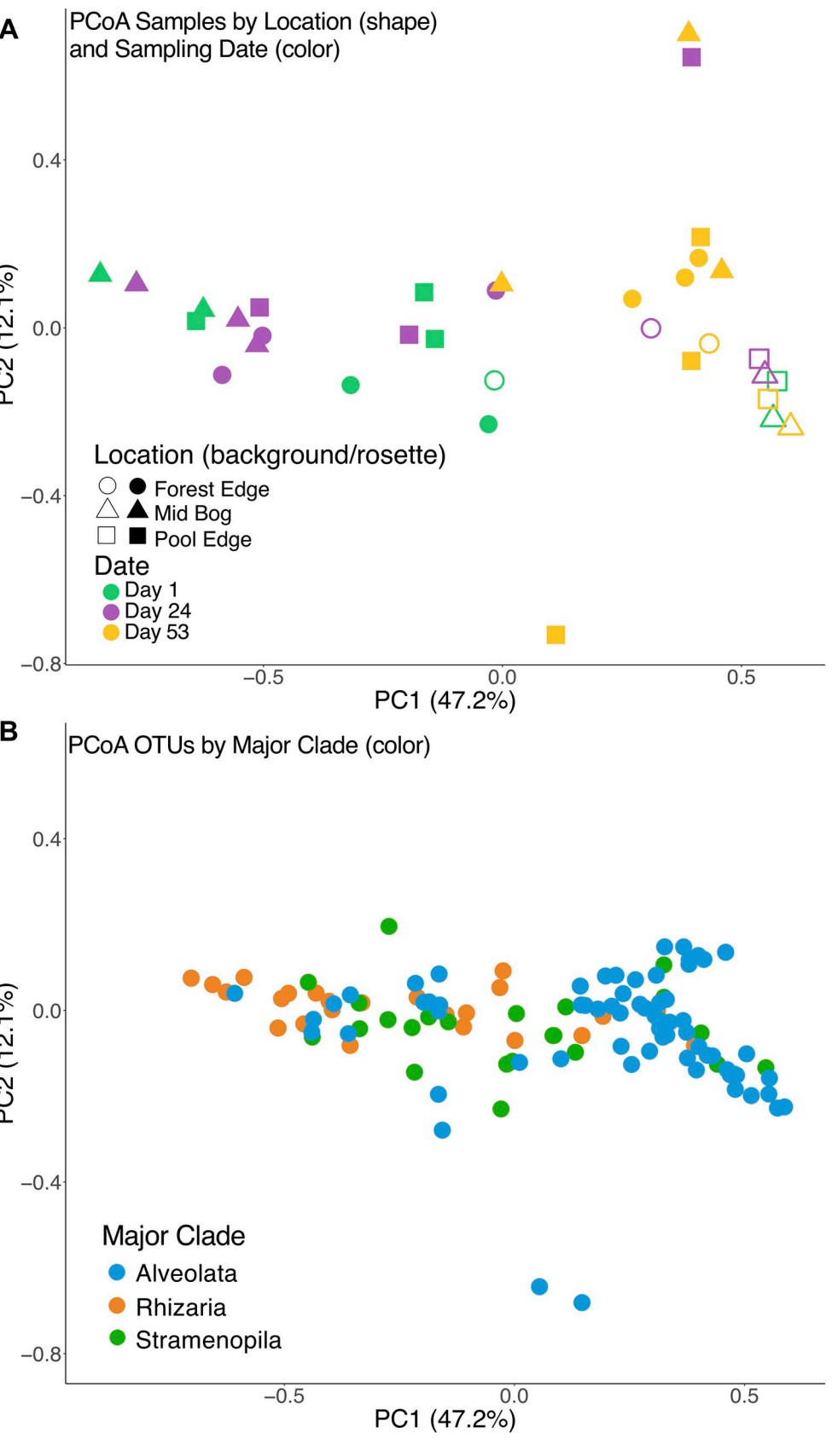

**Fig 4. A**. Principal Coordinate Analysis based on weighted UNIFRAC of communities from repeated sampling along a transect (26m from Forest Edge to Pool Edge) at Hawley Bog (May 6th ("Day 1", green); May 29th ("Day 24", purple); June 27th ("Day 53", yellow)) shows *Sarracenia* SAR communities in late June (Day 53, yellow filled symbols mostly appear on right side of PC1) clustering near the background community (all open symbols). **B**. Principal Coordinate Analysis based on weighted UNIFRAC of OTUs (same ordination space as 4A), showing that the OTUs that differentiate and correspond to the late June samples (the yellow, Day 53 samples on right side of PC1 in 4A) are primarily Alveolate OTUs (denoted in blue). Details on site and OTUs can be found in S1 and S2 Files.

community structure, but with tremendous variation 3) at a single site, SAR communities in wild pitcher plants converge and resemble the background communities as the growing season progresses. Together these data highlight the complexity of microeukaryotic communities associated with plants.

## Richness

Previous work investigating physiological and environmental drivers of protist richness in pitcher plants relied on morphological identification, which makes comparison with molecular studies difficult as richness may be underestimated in morphological studies [50]. In our study, richness measures indicate that in both bromeliads and pitcher plants (both in the field and greenhouse), the inquiline community typically represents only a fraction of the diversity found in nearby standing water (Fig 2). Previous studies have shown that the fluid of pitcher plants and bromeliads is a specialized environment with strong fluctuations of volume, temperature, pH, and chemical composition (reviewed in [10]). These fluctuations make the inquiline environment distinct from the surrounding habitats and likely explain the lower richness observed in the pitcher and bromeliad tanks. This pattern could also be due to limited dispersal of SAR community members into the modified leaves of pitcher plants and bromeliads. Previous work on fungi demonstrated that dispersal ability was a key driver of metacommunities in purple pitcher plants [24].

As we hypothesized, samples from wild pitchers had higher richness than samples from cultivated tropical pitcher plants and bromeliads growing in the Lyman Conservatory (Fig 2). The high richness in wild *Sarracenia* may result from the island-like nature of these pitchers and/or the variation in source populations. We recognize though that interpretation of patterns are confounded by the fact that we sample greenhouse and wild pitchers and different genera, although Bittleston et al. [5] used amplicon sequencing methods to show that *Nepenthes* and *Sarracenia* pitchers grown together developed similar bacterial and eukaryotic inquiline communities. The indoor habitat may have more limited access to protists compared to pitchers in wild habitats, which were sampled across sites in two states, potentially increasing the ability to detect diversity. We did not detect a pattern of increased richness with increased volume as others have ([28]; S2 Fig).

## Study 1: The influence of plant host on community structure

To investigate how host plant characteristic influence protist communities, we sampled from a mix of cultivated and wild phytotelmata forming plants (Table 1). While we detected some differentiation among a subset of samples from bromeliad tanks (i.e. cluster of 7 brown triangles on left side of PC1; Fig 2), there was generally a lack of differentiation between communities from different host species, with samples from bromeliads, *Sarracenia*, and *Nepenthes* overlapping in ordination space (Fig 2). The extensive overlap in SAR communities among hosts could be due to a homogeneous species pool of potentially colonizing protists at the scale of northern New England (i.e. the same protists have equal chance of colonizing bromeliads or pitcher plants because the same protist species/propagules exist at each site). Comparisons of

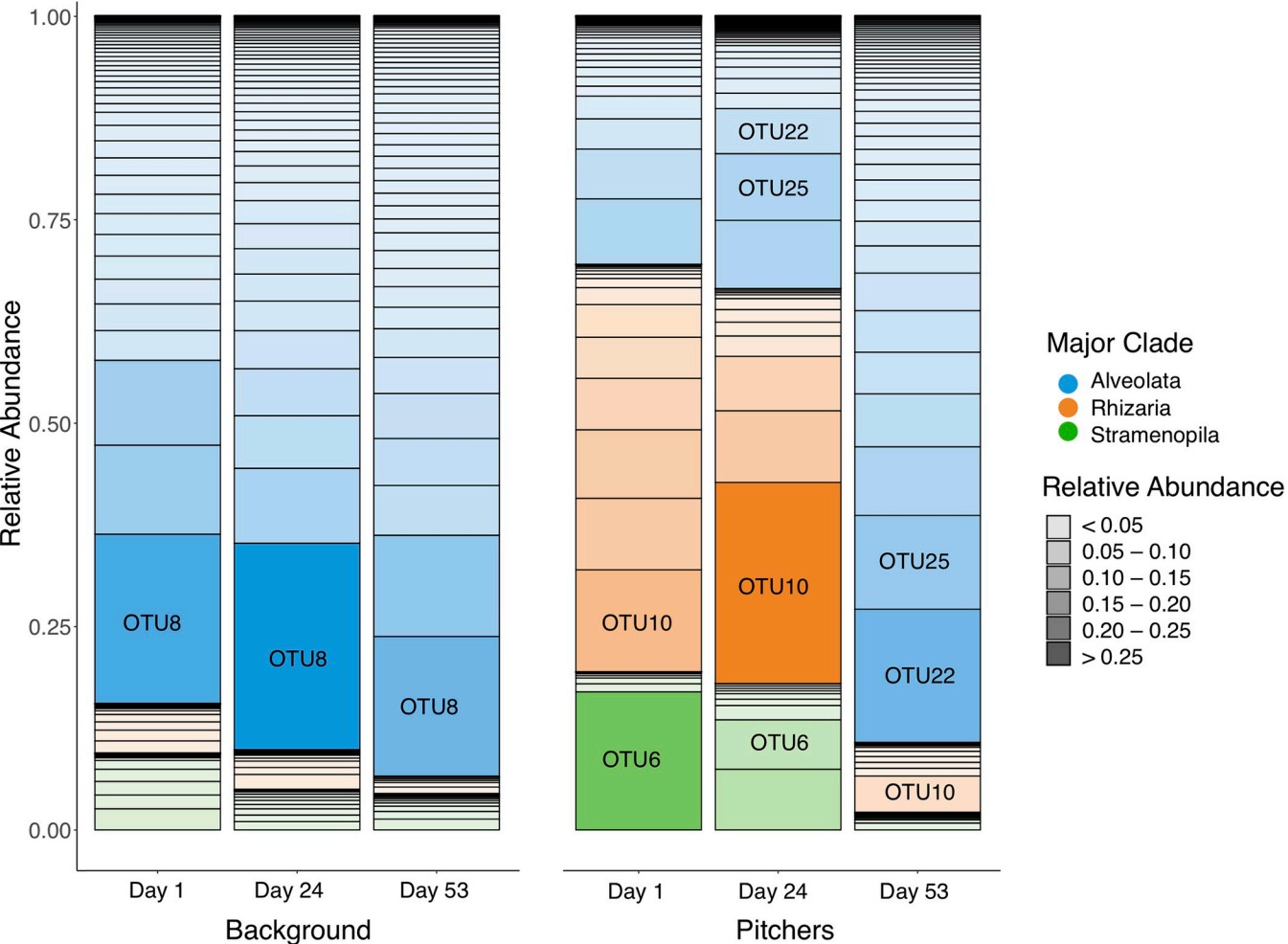

**Fig 5.** The changes in relative abundance of the OTUs from each sampling month separated by sampling type (background bog water (left) vs pitchers (right); OTUs from discussion labeled) shows that the relative abundance of OTUs assigned to Rhizaria declines sharply over the growing season in pitchers, though much less drastically in the background samples. At the same time, the relative abundance of alveolates increases. Stacked barplots represent relative abundance (denoted by bar segment size and shading).

the variation within and between host species raise interesting questions. The variability of communities from *Sarracenia purpurea* from two sites across five dates is understandably high given the range of locations and dates (Fig 2), but contrasts with the almost equal variation detected from bromeliads growing in a single room (i.e. samples from bromeliads span a similar range along PC1 compared to samples from *Sarracenia* in Fig 2). This could be due to the species level morphological differences of tank forming bromeliads as differences in the shape of the tank likely contribute to dispersal patterns and water flow, perhaps leading to the observed variation in SAR communities. Previous work investigating eukaryotic and bacterial communities from wild bromeliads has demonstrated that extensive variation exists within and between bromeliad species [19, 30, 32], supporting our findings of variation in SAR communities within bromeliads found within 3 feet of one another in the Lyman Conservatory.

Varying source communities within the Lyman Conservatory are insufficient to explain the SAR biodiversity patterns in this study. Samples from the tropical pitcher plant genus *Nepenthes* grown in the Lyman Conservatory were distinct from bromeliad samples in our PCoA

(Fig 2), indicating that within the Lyman Conservatory, host plant type influences SAR community structure, though some *Nepenthes* were grown in hanging baskets as opposed to shelf top bromeliad pots. This is also supported by the clustering of SAR communities within two species of *Nepenthes* (Fig 2). In a review of carnivorous pitcher plants, Adlassnig et al. [10] synthesize evidence demonstrating that the *Nepenthes* pitchers secrete digestive enzymes and actively change the pH of their fluid, which may lead to the lower diversity and homogeneity among samples from this genus that we observe.

## Study 2: The influence of season on community structure

The microbial communities present in phytotelmata food webs are useful for understanding ecosystem processes such as colonization and community turnover, as each cavity/pitcher can be considered a unique microcosm that can be sampled at distinct levels including cavities/ pitchers within plants and plants within populations [12, 28]. Similar to previous studies of eukaryotes [6], we detect high variability of eukaryotic microbial communities between pitchers on a single rosette and in a single bog. Repeated monitoring of communities over time in our study found that, counter to our original hypothesis, this variability decreased over the course of the summer in Hawley Bog (i.e. SAR communities within pitchers in late June were more homogeneous than samples in early May). This leads to a number of follow-up hypotheses: in early spring, stochastic colonization events from the background community, as well as differences in pitcher age, lead to diverse communities in pitchers but then as summer progresses, communities are homogenized through dispersal (e.g. rain/wind storms) and/or biotic interactions (e.g. competition, predation) or other ecological and environmental phenomena occur such that, by mid-summer the communities are similar in composition.

Previous research has shown a similar pattern for bacteria and protists where seasonal dynamics can lead to community homogenization over time [51–54]. Bacteria in the phyllosphere have clear seasonal patterns dependent on priority effects whereby the success of one early strain influences the outcome of the entire community [51, 55]. Although we remove fluid containing SAR community members throughout the course of the study, we left behind a portion of the community (in both the water column and biofilm), particularly in the tapering base of each pitcher that likely serves as a refugia for microbial species. Moreover, the stochastic nature of the system means that rainstorms, animal movement, and other factors influence the dispersal of organisms into and out of these systems, making our removal of fluid aligned with the natural processes influencing these communities. Furthermore, the communities in pitcher plants on Day 53 were quite similar to the background communities (Fig 4, yellow filled symbols are clustered with open filled background samples), rather than a novel community cluster representing experimental disturbance. Finally, several studies of the pitcher plant mosquito (*Wyeomyia smithii*) found that densities peaked from late July through September [56, 57] indicating that the phenology of animals in phytotelmata could influence the microbial community temporally. Nevertheless, we recognize that our perturbations may have influenced the observed community changes over time.

For microbial eukaryotes, a competition-colonization model suggests that smaller species will increase population sizes quickly and colonize habitats more rapidly [58–60]. Pitchers in early spring may be dominated by smaller species: indeed, we see Cercozoa (e.g. OTU10, OTU18) dominating, which are likely small flagellate lineages, while larger ciliate species dominate in summer and background samples. Furthermore, as competition increases over the summer, this may lead to higher dispersal that in turn could lead to the homogenizing effect we observe.

## Conclusions

In conclusion, our study demonstrates that host plant type does not always strongly influence phytotelmata protist communities as we find considerable variation in SAR communities within bromeliads, *Sarracenia purpurea*, and *Nepenthes* pitcher plants. We also detect tremendous heterogeneity across space and time, reflecting the stochastic nature of phytotelmata environments. Within a single site over the course of the spring growing season, seasonal dynamics may decrease heterogeneity and point to the importance of repeated temporal sampling. These findings indicate that protists are an integral part of phytotelmata communities and mirror ecosystem level phenomena such as seasonality and species turnover.

## Supporting information

**S1 Fig. Rarefaction analysis of samples from Hawley Bog (where repeated sampling occurred) comparing communities from three pitchers on a single rosette at each location in the bog, and background samples below each rosette.** Communities from background water samples (open symbols) are consistently more species rich than from *Sarracenia* pitchers at Hawley Bog. The background water at the forest edge (light green) is consistently more species rich than the mid bog sites (pink and light blue) along our transect. Error bars are standard deviations.
(TIF)

**S2 Fig. Observed richness (number of OTUs) by pitcher opening diameter and pitcher volume shows no relationship between richness and pitcher morphology.**
(TIF)

**S3 Fig. The PCoA from Fig 2 with 95% data ellipses representing the multivariate t-distribution (solid) and multivariate normal distribution (dashed).**
(TIF)

**S1 File. Sample SRA and collection detail information.**
(CSV)

**S2 File. OTU table with phylogenetic taxonomic assignment and read counts per OTU.**
(CSV)

## Acknowledgments

We are grateful to Ayla Say, Berry Williams, Chase Macpherson, Chrisshara Robinson, Lucinda DeBolt, and Tiannettie McKee for fieldwork and lab assistance. We thank Rabindra Thakur and members of the Katzlab for feedback on early drafts. We extend our appreciation to Janet Atoyan at the University of Rhode Island for running the MiSeq sequencing of our samples.

## Author Contributions

**Conceptualization:** Robin S. Sleith.

**Data curation:** Robin S. Sleith.

**Formal analysis:** Robin S. Sleith, Laura A. Katz.

**Funding acquisition:** Laura A. Katz.

**Investigation:** Robin S. Sleith.

**Methodology:** Robin S. Sleith, Laura A. Katz.

**Project administration:** Robin S. Sleith.

**Resources:** Robin S. Sleith.

**Software:** Robin S. Sleith.

**Supervision:** Laura A. Katz.

**Validation:** Robin S. Sleith.

**Visualization:** Robin S. Sleith.

**Writing – original draft:** Robin S. Sleith.

**Writing – review & editing:** Robin S. Sleith, Laura A. Katz.

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
