## [Decision Letter · Decision Letter 0]

13 Jan 2022

PONE-D-21-40078Illuminating Protist Diversity in Pitcher Plants and Bromeliad TanksPLOS ONE

Dear Dr. Sleith,

Thank you for submitting your manuscript to PLOS ONE. After careful consideration, we feel that it has merit but does not fully meet PLOS ONE’s publication criteria as it currently stands. Therefore, we invite you to submit a revised version of the manuscript that addresses the points raised during the review process.

 The three reviewers have provide detailed comments and recommendations for improvements to the manuscript. Please consider the reviews carefully and pay particular attention to reviewer 3's issue with succession - this needs to be addressed. 

We look forward to receiving your revised manuscript.

Kind regards,

Theodore Raymond Muth

Academic Editor

PLOS ONE

Journal Requirements:

"This work is supported by an NSF grant DEB-1541511 to LAK."

"This work is supported by an NSF grant DEB-1541511 to Laura A. Katz. The funders had no role in study design, data collection and analysis, decision to publish, or preparation of the manuscript."

Reviewers' comments:

Reviewer's Responses to Questions

**Comments to the Author**

1. Is the manuscript technically sound, and do the data support the conclusions?

Reviewer #1: Yes

Reviewer #2: Yes

Reviewer #3: Partly

2. Has the statistical analysis been performed appropriately and rigorously? 

Reviewer #1: Yes

Reviewer #2: Yes

Reviewer #3: Yes

3. Have the authors made all data underlying the findings in their manuscript fully available?

Reviewer #1: Yes

Reviewer #2: Yes

Reviewer #3: Yes

4. Is the manuscript presented in an intelligible fashion and written in standard English?

Reviewer #1: Yes

Reviewer #2: Yes

Reviewer #3: Yes

5. Review Comments to the Author

Reviewer #1: This is a straightforward report of two small-scale studies analyzing the composition of protist communities in pitcher plant tanks, both in different species and different timepoints. It addresses a novel question, executed in a generally technically sound manner, readable, and is a good fit for PLoS ONE. All of my suggestions could feasibly be addressed without additional wetlab experimentation.

One addition that could improve the paper is some additional information within the main text figures on the identity of the OTUs of interest. The introduction indicates that ciliates are dominant in other studies of bromeliad protist communities using different methods, and that bromeliads harbor rare species, but universal primers miss a lot of diversity that SAR primers could capture- three interesting hypotheses. There isn’t much detail given on the taxonomic resolution of these primers, but the final results paragraph and previous Sisson study cited indicates that the sequences are identifiable to some more specific taxonomic level. This information is not necessary for all the OTUs, but would be helpful to add to the dominant OTU highlighted in Fig. 4 and the specialized OTUs shown in figure 5, to put those initial questions into context. Furthermore, several recent studies of protists living on other types of aerial plant tissues have indicated the presence of a large diversity of Cercozoa esp. dominated by Sarcomonads (Ploch et al. 2016, Flues et al. 2018, Walden et al. 2021, Sun et al. 2021), but a few Ciliophora (Colpodea) have been reported to be universal too (Muller and Muller 1970). Some additional OTU identity could help contextualize which patterns you are observing are common to other plant structures vs. novel or specialized to bromeliads. Given the acidity, one would expect the residents to be very different from that of other plants, and the paragraph starting on line 278 indicates this could be true! Paper dois below:

10.1111/jeu.12314

https://doi.org/10.1111/jeu.12503

https://doi.org/10.1111/1462-2920.15385

https://doi.org/10.2307/2423721

https://doi.org/10.1093/femsec/fiab081

The introduction talks a lot about pitcher plant microbiomes, but could benefit from a little more general information on the pitcher plant environments being studied, to be accessible to researchers not in this specific area. Are the tanks thought to be rain-fed or does the water mainly seep in from the surrounding bog? What type of plant exudates feed the microbiome (i.e., sugars, lipids, insecticidal enzymes, acidity?). When one plant has multiple pitchers, do they usually emerge at the same time or do they have different ages? Is the type and amount of insect prey known to affect the microbiome? Finally, what are the phenotypic differences among the species being studied here (volume, habitat, shape, prey range, lids vs. no lids)? This information would help rationalize your hypotheses about species effects and seasonality.

Also in the introduction, there are some references to there being more studies of bacteria and animal communities than of microbial eukaryotes (line 113), meaning protists, but fungal studies are only mentioned in the discussion. Fungi are technically microbial eukaryotes, so please clarify when you are talking about only protist communities.

Line 126- the genetic diversity of the microbial community was similar to the host plant? Unclear phrasing.

-Line 145 “differentiate active community members from cysts” makes it sound like both parts of the community were measured and compared, it would be better rephrased to say the method primarily targets active protists.

Methods:

-Please provide more information about the sequencing beyond that it was MiSeq. Did you sequence paired end reads? 2x300 or 2x150 length? How many total reads? Were reads filtered for quality or length before entering them in the pipeline? Even if some of the QC is in the cited link, it’s best to say briefly what was done.

-line 201 – meaning of “OTU libraries” is unclear.

In figure 3A it would be helpful to indicate somewhere in the figure or legend that open symbols are background samples. Circles indicating confidence intervals and p-values would greatly help demonstrate the finding that Day 53 communities are significantly different.

I really do not understand what Figure 3B is showing. PCA is typically used to differentiate different sample types, but these are all from the same treatment. If the point is that most OTUs are alveolate at this timepoint, isn’t that already shown in figure 4, and what are the principal components for?

Figure 5 is discussed before Figure 3. Figures should be in the order they are discussed in the text.

There are some places in the results where the verb tense suddenly changes from past to present tense (such as line 268). Please check for verb tense consistency.

I would suggest limiting the discussion to be no longer than the length of the results text, which should be enough to synthesize how the results match initial expectations stated in the introduction and extend upon previous work. The recent Walden et al. paper on phenology of protists in aerial plant tissues may be good to mention for context. Also it seems that the findings here (seasonal successional shifts, origin from surrounding environmental microbiota, and specialized/reduced diversity from surrounding microbiota) align well with other emerging paradigms from the leaf and floral ecology field, which may be worth a mention (good overview by Britt Koskella here: https://doi.org/10.1016/j.cub.2020.07.037)

Reviewer #2: In their manuscript “Illuminating Protist Diversity in Pitcher Plants and Bromeliad Tanks”, the authors present a study that used a molecular approach to characterizing eukaryotic microbes (specifically SAR protists) in pitcher plants (Sarracenia and Nepenthes) and tank bromeliads. In the study, the authors sampled inquiline communities from their respective hosts over varying spatial and temporal scales. They found that host plant species affects SAR community structure and that there is a large amount of variability between communities within the same host plant. They also found that Sarracenia pitcher plants from the same bog are more diverse and heterogenous earlier in the growing season than later in the growing season. I enjoyed reading this manuscript immensely. The methods are rigorous and stand out from previous work that characterizes protists in phytotelma primarily based on morphology. Overall, I believe that the manuscript has merit for publication following a round of revisions from the authors. I have made some specific suggestions below.

Thank you for a fun read,

Erica Holdridge

NSF Postdoctoral Research Fellow

Boise State University

1. Although the authors have provided a nice table (Table 1), I still have a hard time understanding the sampling structure of their study. This could use quite a bit of clarification in the text of the manuscript. In particular, there are 3 samples from each bromeliad species taken across 2 dates. Are these from 3 different plants? How many were sampled on each date? etc. The same questions apply for the Nepenthes sampled. There were also 17 Sarracenia samples from Hawley Bog separately from the temporal study. Were these from a separate part of the bog than the Sarracenia that were used in the temporal study?

2. I can’t help but wonder if your Sarracenia samples are so variable (e.g. Figure 2B) because (a) you simply have more samples taken over a wider range of time points and (b) they’re taken from the field rather than a greenhouse.

3. Lines 341-346: Are Nepenthes at Lyman Conservatory grown in hanging baskets? If so, this may be one reason why bromeliads and Nepenthes from the same greenhouse have distinct communities (although I do believe that most of it does have to do with host identity).

4. Do you notice more mosquitoes (particularly W. smithii) in Hawley bog later in the growing season? It’s possible that they are aiding in dispersal and could explain some of the homogeneity you see later in the growing season. You do mention “animal movement” on lines 371-372, which may be a reference to this idea.

Minor comments:

• lines 77-78: “Much less is known about freshwater habitats” to “Much less is known about SAR in freshwater habitats”

• lines 92-93: “host available” to “host-available”

• line 163: “was collected” to “were collected”

• line 203: “SWARM v2 d = 1” to “SWARM v2 with the parameter d = 1”

• line 212: “weighted and unweighted (for relative abundance)” to “weighted (relative abundance) and unweighted (presence/absence)”

• lines 231-232: How many OTUs exactly?

• lines 253-255: The first sentence of this section is a fragment.

• line 267: You discuss Figure 5 before Figure 4; they should probably be reordered so they are organized by the order in which they are discussed

• line 286: OTU8 seems to be most abundant in the background bog water (Figure 4) – what is the taxonomic assignment of that OUT?

• line 342: This is the first place where you clarify that you mean Nepenthes when you say “tropical pitcher plants”. I know that this is the common name but other readers may not, so this should be clarified earlier in the manuscript.

• line 355: This would be a great place to site Aaron Ellison and Nick Gotelli’s new book “Scaling Ecology with a Model System” (2021)

Reviewer #3: The manuscript by Sleigh and Katz investigates the eukaryotic diversity within the inquiline communities of pitcher plants by using a taxon-focused approach. The technique they used is very timely and needed because eukaryotic diversity is understudied in general and especially in pitcher plant systems. When eukaryotic diversity is studied, it is usually with broad primers that can underestimate our knowledge about protist diversity because of the overabundance of insect DNA that can also be captured. Overall, the manuscript was very well written and there was a very indepth literature search. My main comments are that more information is needed in the methods section, and I am not fully comfortable with considering this a study that addresses Succession. With most of the inquiline community removed at each sampling point, this study seems to address more the resilience of the community after a disturbance, and the role of stochastic events and dispersal, instead of internal biotic interactions, on protist diversity throughout time, than a proper assessment of Succession. The study did definitely investigate the difference in protist composition across the different locations in the bog, but their question about Succession may need to be re-thought.

Main comments in Method Section:

In the Methods section of Study 2, additional information is needed in order for a better understanding of what was done. Specifically,

1) Were new pitchers selected on each of the rosettes at the beginning of the field season? This would allow for an accurate assessment of early succession.

2) It is unclear in the Methods if the three pitchers were sampled on each of the three sampling dates, or if one pitcher represented a specific sampling date and the replicate was on the other two rosettes in the section (i.e., Pitcher 1 = May 6, Pitcher 2 = May 29, Pitcher 3 = June 27). This information is stated in the Results section (Line 253-255) and should be moved up to the Methods for clarity.

3) For a better assessment of the methods and results, it would also be helpful to know how much volume was in each pitcher during each sampling time and the distance between each plant in Forest edge versus open bog, for example.

Main comment regarding Succession:

On Line 179-181, the authors state that a maximum of 25mL of fluid was removed and fluid was concentrated onto filters. If each pitcher was sampled three times (the three time periods), removing up to 25mL on each time period would be an exhaustive sampling, which would not capture protist succession through time if the community was removed almost completely during each sampling and needed to reset. To follow succession properly, it would be better to either exhaustively remove the volume of one pitcher on each of 3 replicate rosettes. This would represent one time period, replicated 3 times within 1 section of the bog, or, to take a smaller volume from each pitcher during the three sampling events to prevent a disturbance.

My concern about the large volume that was removed each time continues in Lines 361-365 of the Discussion. These follow-up questions are indeed important, but I just cannot see how biotic interactions will play a role in the authors’ sampling method if they removed most of the volume in the leaf each time. The internal biotic interactions were basically reset at each time point, making it instead a study that addressed the role of stochastic events and dispersal at different time points throughout the growing season, and a study that tests the resilience of the community after such a large disturbance each time. (I am happy to see though that the authors recognize this potentially large perturbation of removing almost all the volume during each sampling point (Line 374-375) and their results do indeed match what is found with microscopic counts of protists in the Sarracenia system, with flagellates dominating at the beginning of succession and ciliates being more abundant later in the season).

Additional comments:

- Why were OTUs used instead of ASVs?

- Figure 2. It would be helpful to visualize which of the Sarracenia samples came from which site.

- Figure 3A. Labelling the Date as a circle is confusing with the Forest Edge circle. Additionally, it is confused what “Pool Edge” means.

- Table 1. Minor detail, but signify that the dates are Month/Day.

6. PLOS authors have the option to publish the peer review history of their article (what does this mean?). If published, this will include your full peer review and any attached files.

Reviewer #1: No

Reviewer #2: **Yes: **Erica Holdridge

Reviewer #3: No

---

## [Author Response · Author response to Decision Letter 0]

7 Mar 2022

We thank the reviewers for their helpful comments, which we have addressed in detail below. We believe the manuscript is now substantially strengthened and more accessible to the broad readership.

****

Reviewer 1

One addition that could improve the paper is some additional information within the main text figures on the identity of the OTUs of interest. The introduction indicates that ciliates are dominant in other studies of bromeliad protist communities using different methods, and that bromeliads harbor rare species, but universal primers miss a lot of diversity that SAR primers could capture- three interesting hypotheses. There isn’t much detail given on the taxonomic resolution of these primers, but the final results paragraph and previous Sisson study cited indicates that the sequences are identifiable to some more specific taxonomic level. This information is not necessary for all the OTUs, but would be helpful to add to the dominant OTU highlighted in Fig. 4 and the specialized OTUs shown in figure 5, to put those initial questions into context. Furthermore, several recent studies of protists living on other types of aerial plant tissues have indicated the presence of a large diversity of Cercozoa esp. dominated by Sarcomonads (Ploch et al. 2016, Flues et al. 2018, Walden et al. 2021, Sun et al. 2021), but a few Ciliophora (Colpodea) have been reported to be universal too (Muller and Muller 1970). Some additional OTU identity could help contextualize which patterns you are observing are common to other plant structures vs. novel or specialized to bromeliads. Given the acidity, one would expect the residents to be very different from that of other plants, and the paragraph starting on line 278 indicates this could be true! Paper dois below:

REPLY: Thank you for the suggestion, we agree that we could do more to teach about the biology of the organisms. To this end, we have added some additional information to the results and point to the OTU table in the supplement which has all taxa identified to species with the corresponding GenBank number of the closest reference taxon in the tree-based taxonomy assignment.

The introduction talks a lot about pitcher plant microbiomes, but could benefit from a little more general information on the pitcher plant environments being studied, to be accessible to researchers not in this specific area. Are the tanks thought to be rain-fed or does the water mainly seep in from the surrounding bog? What type of plant exudates feed the microbiome (i.e., sugars, lipids, insecticidal enzymes, acidity?). When one plant has multiple pitchers, do they usually emerge at the same time or do they have different ages? Is the type and amount of insect prey known to affect the microbiome? Finally, what are the phenotypic differences among the species being studied here (volume, habitat, shape, prey range, lids vs. no lids)? This information would help rationalize your hypotheses about species effects and seasonality.

REPLY: We have added additional information to the introduction that explains more pitcher biology and ecology. While we would love to address all of the questions raised here we believe this would be beyond the scope of the introduction to a paper investigating microbial communities and instead point to the literature and some helpful review papers.

Also in the introduction, there are some references to there being more studies of bacteria and animal communities than of microbial eukaryotes (line 113), meaning protists, but fungal studies are only mentioned in the discussion. Fungi are technically microbial eukaryotes, so please clarify when you are talking about only protist communities.

REPLY: We have clarified where fungi fit into the phytotelmata literature.

Line 126- the genetic diversity of the microbial community was similar to the host plant? Unclear phrasing.

REPLY: We have reworded the sentence to clarify.

Line 145 “differentiate active community members from cysts” makes it sound like both parts of the community were measured and compared, it would be better rephrased to say the method primarily targets active protists.

REPLY: We have rephrased this sentence to more accurately reflect our methods.

Please provide more information about the sequencing beyond that it was MiSeq. Did you sequence paired end reads? 2x300 or 2x150 length? How many total reads? Were reads filtered for quality or length before entering them in the pipeline? Even if some of the QC is in the cited link, it’s best to say briefly what was done.

REPLY: We have added the requested information to the methods.

line 201 – meaning of “OTU libraries” is unclear.

REPLY: We have added language to clarify the sentence.

In figure 3A it would be helpful to indicate somewhere in the figure or legend that open symbols are background samples. Circles indicating confidence intervals and p-values would greatly help demonstrate the finding that Day 53 communities are significantly different.

REPLY: The key of 3A indicates (background/rosette) for open vs. filled and we have added text to the legend. The addition of confidence intervals makes the figure too difficult to interpret but the Adonis test reported in the results demonstrates that the relationship is significant.

I really do not understand what Figure 3B is showing. PCA is typically used to differentiate different sample types, but these are all from the same treatment. If the point is that most OTUs are alveolate at this timepoint, isn’t that already shown in figure 4, and what are the principal components for?

REPLY: This is the same PCoA as Figure 3B but instead of plotting samples, we are plotting the taxa (OTUs) in the same PCoA space, which demonstrates which taxa (or groups of taxa) are driving the differentiation of samples we see in 3A. We have added text to the legend that helps explain this better.

Figure 5 is discussed before Figure 3. Figures should be in the order they are discussed in the text.

REPLY: Apologies, we have fixed the figure order.

There are some places in the results where the verb tense suddenly changes from past to present tense (such as line 268). Please check for verb tense consistency.

REPLY: We have checked verb tense consistency.

I would suggest limiting the discussion to be no longer than the length of the results text, which should be enough to synthesize how the results match initial expectations stated in the introduction and extend upon previous work. The recent Walden et al. paper on phenology of protists in aerial plant tissues may be good to mention for context. Also it seems that the findings here (seasonal successional shifts, origin from surrounding environmental microbiota, and specialized/reduced diversity from surrounding microbiota) align well with other emerging paradigms from the leaf and floral ecology field, which may be worth a mention (good overview by Britt Koskella here: https://doi.org/10.1016/j.cub.2020.07.037)

REPLY: Thank you, we have received other feedback that requested more discussion, and will work with the editor to reconcile this. We appreciate the suggestions and have incorporated these references.

Reviewer 2

1. Although the authors have provided a nice table (Table 1), I still have a hard time understanding the sampling structure of their study. This could use quite a bit of clarification in the text of the manuscript. In particular, there are 3 samples from each bromeliad species taken across 2 dates. Are these from 3 different plants? How many were sampled on each date? etc. The same questions apply for the Nepenthes sampled. There were also 17 Sarracenia samples from Hawley Bog separately from the temporal study. Were these from a separate part of the bog than the Sarracenia that were used in the temporal study?

REPLY: We thank the reviewer for this question and we have added text to the methods clarifying the sampling design.

2. I can’t help but wonder if your Sarracenia samples are so variable (e.g. Figure 2B) because (a) you simply have more samples taken over a wider range of time points and (b) they’re taken from the field rather than a greenhouse.

REPLY: In the discussion we mention the high variation of Sarracenia and expect this high variation given the samples are coming from different locations and dates. Also, these pitchers serve as islands in their environments, leading to high variability. What we did not expect was that the variation of bromeliads was nearly as great as Sarracenia. We have clarified wording in the discussion regarding this.

3. Lines 341-346: Are Nepenthes at Lyman Conservatory grown in hanging baskets? If so, this may be one reason why bromeliads and Nepenthes from the same greenhouse have distinct communities (although I do believe that most of it does have to do with host identity).

REPLY: Some Nepenthes are in hanging baskets and some are in pots on shelves at the same height as the bromeliads. This is a possible difference and we have added it to the discussion.

4. Do you notice more mosquitoes (particularly W. smithii) in Hawley bog later in the growing season? It’s possible that they are aiding in dispersal and could explain some of the homogeneity you see later in the growing season. You do mention “animal movement” on lines 371-372, which may be a reference to this idea.

REPLY: Unfortunately, we do not have quantitative counts of mosquito larvae from these samples, but yes, this is a possibility. We have added a discussion of mosquito activity that incorporates this idea.

lines 77-78: “Much less is known about freshwater habitats” to “Much less is known about SAR in freshwater habitats”

REPLY: We have made this correction. 

lines 92-93: “host available” to “host-available”

REPLY: We have made this correction. 

line 163: “was collected” to “were collected”

REPLY: We have made this correction. 

line 203: “SWARM v2 d = 1” to “SWARM v2 with the parameter d = 1”

REPLY: We have made this correction. 

line 212: “weighted and unweighted (for relative abundance)” to “weighted (relative abundance) and unweighted (presence/absence)”

REPLY: We have made this correction. 

lines 231-232: How many OTUs exactly?

REPLY: We have added OTU number to this sentence.

lines 253-255: The first sentence of this section is a fragment.

REPLY: We have made this correction. 

line 267: You discuss Figure 5 before Figure 4; they should probably be reordered so they are organized by the order in which they are discussed

REPLY: We have fixed the figure order. 

line 286: OTU8 seems to be most abundant in the background bog water (Figure 4) – what is the taxonomic assignment of that OUT?

REPLY: We have added information on OTU8 to the results.

line 342: This is the first place where you clarify that you mean Nepenthes when you say “tropical pitcher plants”. I know that this is the common name but other readers may not, so this should be clarified earlier in the manuscript.

REPLY: We have made this correction. 

line 355: This would be a great place to site Aaron Ellison and Nick Gotelli’s new book “Scaling Ecology with a Model System” (2021)

REPLY: Thank you for the suggestion, we have added the reference

REVIEWER 3

Reviewer #3: The manuscript by Sleigh and Katz investigates the eukaryotic diversity within the inquiline communities of pitcher plants by using a taxon-focused approach. The technique they used is very timely and needed because eukaryotic diversity is understudied in general and especially in pitcher plant systems. When eukaryotic diversity is studied, it is usually with broad primers that can underestimate our knowledge about protist diversity because of the overabundance of insect DNA that can also be captured. Overall, the manuscript was very well written and there was a very indepth literature search. My main comments are that more information is needed in the methods section, and I am not fully comfortable with considering this a study that addresses Succession. With most of the inquiline community removed at each sampling point, this study seems to address more the resilience of the community after a disturbance, and the role of stochastic events and dispersal, instead of internal biotic interactions, on protist diversity throughout time, than a proper assessment of Succession. The study did definitely investigate the difference in protist composition across the different locations in the bog, but their question about Succession may need to be re-thought.

REPLY: We appreciate the point regarding succession vs. disturbance and have further addressed the difficulty in untangling succession and disturbance in the discussion.

1) Were new pitchers selected on each of the rosettes at the beginning of the field season? This would allow for an accurate assessment of early succession.

REPLY: Pitchers die back after each season and only “new growth” pitchers were selected. That being said, the pitchers were already open by the start time of the study. 

2) It is unclear in the Methods if the three pitchers were sampled on each of the three sampling dates, or if one pitcher represented a specific sampling date and the replicate was on the other two rosettes in the section (i.e., Pitcher 1 = May 6, Pitcher 2 = May 29, Pitcher 3 = June 27). This information is stated in the Results section (Line 253-255) and should be moved up to the Methods for clarity.

REPLY: We have moved this information to the methods section for clarity.

3) For a better assessment of the methods and results, it would also be helpful to know how much volume was in each pitcher during each sampling time and the distance between each plant in Forest edge versus open bog, for example.

REPLY: We have added the volume of fluid in each pitcher to the Supplementary Table S1. The rosettes were spaced along a 26m transect with approximately 8 meters between rosettes.

On Line 179-181, the authors state that a maximum of 25mL of fluid was removed and fluid was concentrated onto filters. If each pitcher was sampled three times (the three time periods), removing up to 25mL on each time period would be an exhaustive sampling, which would not capture protist succession through time if the community was removed almost completely during each sampling and needed to reset. To follow succession properly, it would be better to either exhaustively remove the volume of one pitcher on each of 3 replicate rosettes. This would represent one time period, replicated 3 times within 1 section of the bog, or, to take a smaller volume from each pitcher during the three sampling events to prevent a disturbance.

REPLY: We realize that removing liquid will influence the community and impact the measurement of succession, and we discuss this in lines 400-410. We note that the dynamics we observe are in line with other studies of microbial community succession. We are careful to mention disturbance and argue that our results are put in proper context of our sampling design.

My concern about the large volume that was removed each time continues in Lines 361-365 of the Discussion. These follow-up questions are indeed important, but I just cannot see how biotic interactions will play a role in the authors’ sampling method if they removed most of the volume in the leaf each time. The internal biotic interactions were basically reset at each time point, making it instead a study that addressed the role of stochastic events and dispersal at different time points throughout the growing season, and a study that tests the resilience of the community after such a large disturbance each time. (I am happy to see though that the authors recognize this potentially large perturbation of removing almost all the volume during each sampling point (Line 374-375) and their results do indeed match what is found with microscopic counts of protists in the Sarracenia system, with flagellates dominating at the beginning of succession and ciliates being more abundant later in the season).

REPLY: The morphology of these pitcher plants is such that it was impossible to remove all the fluid from the plant. While removal of the liquid is certainly a disturbance, the size of the organisms being studied meant that many cells likely remained, and likely in similar proportions as the rest of the fluid. These plants are also subject to rain (possibly flooding/diluting the community) and wind (loss of volume), and so their natural state is quite stochastic. 

Why were OTUs used instead of ASVs?

REPLY: OTUs were used to align this study with previous work using the SAR specific primers. We agree that ASVs have utility but can be difficult to interpret for groups such as ciliates that have intra-individual variability of rDNA. Moreover, unpublished data from our lab have shown substantial problems with DADA2 when applied to microeukaryotes: it is not parameterized to capture the length variation found among microeukaryotes.

Figure 2. It would be helpful to visualize which of the Sarracenia samples came from which site.

REPLY: We did try this initially but the many shapes made the figure too difficult to interpret. We now point our reader to supplement that includes these details.

Figure 3A. Labelling the Date as a circle is confusing with the Forest Edge circle. Additionally, it is confused what “Pool Edge” means.

REPLY: We have added text to the legend to clarify the shapes and the meaning of pool edge.

Table 1. Minor detail, but signify that the dates are Month/Day.

REPLY: We have made this correction.

---

## [Decision Letter · Decision Letter 1]

13 Apr 2022

PONE-D-21-40078R1Illuminating Protist Diversity in Pitcher Plants and Bromeliad TanksPLOS ONE

Dear Dr. Sleith,

Thank you for submitting your manuscript to PLOS ONE. After careful consideration, we feel that it has merit but does not fully meet PLOS ONE’s publication criteria as it currently stands. Therefore, we invite you to submit a revised version of the manuscript that addresses the points raised during the review process.

 In the revised (2nd revision) manuscript, please address the continued concerns of reviewer three. There remains a need for clarification on the age of the pitcher plants sampled and the possible sampling from pitchers that lasted over the winter from an earlier season. If there is a reason that the pitcher plants in this study are maturing earlier than expected by the third reviewer, please make that clear. Please address the question regarding dipteran larvae remaining in the pitcher plants.

We look forward to receiving your revised manuscript.

Kind regards,

Theodore Raymond Muth

Academic Editor

PLOS ONE

Journal Requirements:

Reviewers' comments:

Reviewer's Responses to Questions

**Comments to the Author**

1. If the authors have adequately addressed your comments raised in a previous round of review and you feel that this manuscript is now acceptable for publication, you may indicate that here to bypass the “Comments to the Author” section, enter your conflict of interest statement in the “Confidential to Editor” section, and submit your "Accept" recommendation.

Reviewer #1: (No Response)

Reviewer #2: All comments have been addressed

Reviewer #3: (No Response)

2. Is the manuscript technically sound, and do the data support the conclusions?

Reviewer #1: Yes

Reviewer #2: Yes

Reviewer #3: Partly

3. Has the statistical analysis been performed appropriately and rigorously? 

Reviewer #1: Yes

Reviewer #2: Yes

Reviewer #3: Yes

4. Have the authors made all data underlying the findings in their manuscript fully available?

Reviewer #1: Yes

Reviewer #2: Yes

Reviewer #3: Yes

5. Is the manuscript presented in an intelligible fashion and written in standard English?

Reviewer #1: Yes

Reviewer #2: Yes

Reviewer #3: Yes

6. Review Comments to the Author

Reviewer #1: I feel that the authors responded satisfactorily to the previous comments and that the manuscript meets PLoS ONE review criteria.

Reviewer #2: The authors have revised their manuscript, “Illuminating Protist Diversity in Pitcher Plants and Bromeliad Tanks”, following comments from myself and two other reviewers. The revised manuscript adequately addresses all of my previous comments and I do not have any additional comments. Great job handling all of the (sometimes conflicting) comments and thanks for a fun read!

Reviewer #3: Thank you very much for the revisions to the manuscript. I commend the authors for the work they did. Overall, I find the study interesting and timely. Eukaryotic diversity is understudied in general, and especially in pitcher plant systems. I still, however, have concerns about the methods in Study 2. My concerns do not mean that there is a problem with the data, but rather in the question that is being addressed (Succession). My reasons for this are detailed below:

1) Standardization of pitcher leaf age:

Sampling in the beginning of May in Massachusetts seems very early in the growing season for Sarracenia purpurea at this latitude. It is thus surprising that there were new leaves/pitchers available on the rosettes (especially 3 new leaves per rosette). More surprising is that the new leaves that were used in the study were open and filled with large volumes of water (as shown in the supplemental data), unless several large rainstorms occurred just before sampling. I just can’t imagine that the new pitchers (from that season) would be large enough to hold such a large volume at this time period.

Not all pitchers die back each year. Instead, they overwinter, with the snow protecting them. There will, of course, be some old pitchers from several seasons in the past that have died back (the pitchers closer to the ground), but the middle of the rosette will have pitchers from last season that have survived the winter as well as the new pitchers that are growing in the current season. It can sometimes be difficult to determine last season’s pitchers from new growth pitchers if the new growth pitchers are ~4 weeks or older. I am therefore uncertain if the authors used pitchers from that season or overwintered pitchers. For the study to examine Succession, the age and opening time of the pitchers needed to be better standardized. This can be done by selecting very new (fresh, flexible green leaves that had just opened) pitchers. The first sampling event should have then been done several weeks after they opened, so that there was time for the pitchers to fill with water.

Study 2 still informs us about the inquiline community that inhabits pitchers, but without the initial standardization, it acts more of a snapshot at each point in the season, and not succession.

Apologies if my phenology is incorrect. Addition of field site information (flowering time, new leaf production, precipitation) should be added to the manuscript.

2) Dipteran larvae in the inquiline community:

The large volume of water that was removed from each pitcher during sampling would also remove any of the dipteran larvae that are essential for dynamics of the inquiline community. Some (the midge and some of the mosquito larvae) could be down at the bottom of the leaf near the decomposing insects, so it is likely that those ones could remain in the community after each sampling. It is unfortunate that the authors did not record midge and mosquito larvae during the sampling so that the authors have an idea of how many were lost from the community during each sampling. Were there any mosquito larvae stuck on the filter after the inquiline community was filtered?

Along those lines, a better reference than Rango 1999 for seasonal Wyeomyia dynamics would be to use the Harvard Forest Data Archives. These archives contain data from the authors’ field site and/or sites in the region and have information about the seasonal dynamics of the inquiline mosquito larvae in the system. https://harvardforest1.fas.harvard.edu/exist/apps/datasets/showData.html?id=hf193

It very well might be that the mosquito larvae densities do not peak until mid-summer, and therefore are not a part of the May-June sampling that occurred in this study. In the lower latitude part of Sarracenia purpurea’s distribution (e.g., Florida) the mosquito larvae are the most abundant as soon as the leaf opens. If that is the case in this study site as well, then the mosquito density would be greatly impacted. If overwintered leaves were indeed used for this study, there could be mosquito larvae present because they were in diapause in the leaves during the winter.

3) Disturbance by removing large volumes of water during sampling: a large volume was taken from each community during each sampling time period in Study 2. The authors do replace the volume they removed with the supernatant, but this is still a large, open niche space. I agree that there will be some microbes that remain at the bottom of the leaf during each sampling time period, but the community is undergoing a great disturbance during each sampling period, with diversity potentially being diluted each time.

In summary, I do think this study has accomplished a lot of things and is interesting, it is just that there are some potential issues in the methods which will affect the focus of the paper as it currently is written. In general, I suggest that the authors be careful with the word “succession” in this study. Instead, I suggest that the authors use “seasonal” dynamics or something along those lines.

7. PLOS authors have the option to publish the peer review history of their article (what does this mean?). If published, this will include your full peer review and any attached files.

Reviewer #1: No

Reviewer #2: **Yes: **Erica Holdridge

Reviewer #3: No

---

## [Author Response · Author response to Decision Letter 1]

15 Jun 2022

We thank the reviewers for their helpful comments, which we have addressed in detail below. 

****

Reviewer #3:

Thank you very much for the revisions to the manuscript. I commend the authors for the work they did. Overall, I find the study interesting and timely. Eukaryotic diversity is understudied in general, and especially in pitcher plant systems. I still, however, have concerns about the methods in Study 2. My concerns do not mean that there is a problem with the data, but rather in the question that is being addressed (Succession). My reasons for this are detailed below:

REPLY: As a result of the comments of this reviewer, we have further clarified the language in our manuscript including reframing successional patterns as seasonal change to the community over time. 

1) Standardization of pitcher leaf age:

Sampling in the beginning of May in Massachusetts seems very early in the growing season for Sarracenia purpurea at this latitude. It is thus surprising that there were new leaves/pitchers available on the rosettes (especially 3 new leaves per rosette). More surprising is that the new leaves that were used in the study were open and filled with large volumes of water (as shown in the supplemental data), unless several large rainstorms occurred just before sampling. I just can’t imagine that the new pitchers (from that season) would be large enough to hold such a large volume at this time period.

Not all pitchers die back each year. Instead, they overwinter, with the snow protecting them. There will, of course, be some old pitchers from several seasons in the past that have died back (the pitchers closer to the ground), but the middle of the rosette will have pitchers from last season that have survived the winter as well as the new pitchers that are growing in the current season. It can sometimes be difficult to determine last season’s pitchers from new growth pitchers if the new growth pitchers are ~4 weeks or older. I am therefore uncertain if the authors used pitchers from that season or overwintered pitchers. For the study to examine Succession, the age and opening time of the pitchers needed to be better standardized. This can be done by selecting very new (fresh, flexible green leaves that had just opened) pitchers. The first sampling event should have then been done several weeks after they opened, so that there was time for the pitchers to fill with water.

REPLY: We thank the reviewer for these comments. We have reviewed field photos and believe it is possible that pitchers from the previous season were sampled, given how early in the season it was, and the size of some of the pitchers. We now clarify that the pitchers we sample may be of different generations, and that seasonality refers to turn over of the microbial community from spring through summer. We have also added that variance among spring pitchers might be due to the age of individual pitchers (i.e. those that overwintered vs new growth).

Study 2 still informs us about the inquiline community that inhabits pitchers, but without the initial standardization, it acts more of a snapshot at each point in the season, and not succession.

REPLY: We believe that our improved language has clarified this issue.

Apologies if my phenology is incorrect. Addition of field site information (flowering time, new leaf production, precipitation) should be added to the manuscript.

REPLY: We agree that to understand how the age and phenology of the pitcher plant impacts the inquiline community (e.g. Armitage 2017) requires repeated sampling of pitchers with standardized ages. However, our current study is interested in how the microbial community changes during the spring growing season, irrespective of pitcher age (e.g. Buosi et al. 2015). Pitcher age may be a factor in structuring these communities, but our current study demonstrates that seasonality tends to be a more important driver, given that communities in leaves of different ages converge by the end of the sampling window (Figure 4), as opposed to communities being structured by pitcher age. Nevertheless, to avoid giving a false impression we have removed the term succession when discussing our results and simply use the term seasonal changes. We have also clarified the age of pitchers and sampling strategy as well as site information in the Methods and Discussion.

2) Dipteran larvae in the inquiline community:

The large volume of water that was removed from each pitcher during sampling would also remove any of the dipteran larvae that are essential for dynamics of the inquiline community. Some (the midge and some of the mosquito larvae) could be down at the bottom of the leaf near the decomposing insects, so it is likely that those ones could remain in the community after each sampling. It is unfortunate that the authors did not record midge and mosquito larvae during the sampling so that the authors have an idea of how many were lost from the community during each sampling. Were there any mosquito larvae stuck on the filter after the inquiline community was filtered?

Along those lines, a better reference than Rango 1999 for seasonal Wyeomyia dynamics would be to use the Harvard Forest Data Archives. These archives contain data from the authors’ field site and/or sites in the region and have information about the seasonal dynamics of the inquiline mosquito larvae in the system. https://harvardforest1.fas.harvard.edu/exist/apps/datasets/showData.html?id=hf193

It very well might be that the mosquito larvae densities do not peak until mid-summer, and therefore are not a part of the May-June sampling that occurred in this study. In the lower latitude part of Sarracenia purpurea’s distribution (e.g., Florida) the mosquito larvae are the most abundant as soon as the leaf opens. If that is the case in this study site as well, then the mosquito density would be greatly impacted. If overwintered leaves were indeed used for this study, there could be mosquito larvae present because they were in diapause in the leaves during the winter.

REPLY: We appreciate this insight and have looked at the Ellison & Gotelli dataset. The hf193-04: food web structure data show that for Hawley Bog (as well as Molly Bog) the density for Wyeomyia peaks in August and September (with some detections in July, and very few in June). We have added this information to the discussion.

3) Disturbance by removing large volumes of water during sampling: a large volume was taken from each community during each sampling time period in Study 2. The authors do replace the volume they removed with the supernatant, but this is still a large, open niche space. I agree that there will be some microbes that remain at the bottom of the leaf during each sampling time period, but the community is undergoing a great disturbance during each sampling period, with diversity potentially being diluted each time.

In summary, I do think this study has accomplished a lot of things and is interesting, it is just that there are some potential issues in the methods which will affect the focus of the paper as it currently is written. In general, I suggest that the authors be careful with the word “succession” in this study. Instead, I suggest that the authors use “seasonal” dynamics or something along those lines.

REPLY: The volume of fluid removed does represent a disturbance to the system, and we have added additional text addressing this issue in lines 388-398 of the Discussion. The final sampling window shows that communities in the pitcher plants on Day 53 were quite similar to the background communities (Figure 4, yellow filled symbols are clustered with open fill bog water samples). If we were diluting diversity and changing the community composition drastically, we would expect the communities to be quite unique and would expect samples from Day 53 to cluster on their own near no natural samples. We contend that the dynamic/stochastic nature of the pitcher system is such that the disturbance created by our sampling is not beyond the scope of what is possible in nature, and therefore our results are a useful addition in the study of microbial eukaryote communities. We hope that this study inspires others to track additional communities using other experimental methods.

---

## [Editor Report · Decision Letter 2]

20 Jun 2022

Illuminating Protist Diversity in Pitcher Plants and Bromeliad Tanks

PONE-D-21-40078R2

Dear Dr. Sleith,

We’re pleased to inform you that your manuscript has been judged scientifically suitable for publication and will be formally accepted for publication once it meets all outstanding technical requirements.

Kind regards,

Theodore Raymond Muth

Academic Editor

PLOS ONE
---

## [Editor Report · Acceptance letter]

30 Jun 2022

PONE-D-21-40078R2 

Illuminating Protist Diversity in Pitcher Plants and Bromeliad Tanks 

Dear Dr. Sleith:

I'm pleased to inform you that your manuscript has been deemed suitable for publication in PLOS ONE. Congratulations! Your manuscript is now with our production department. 

Kind regards, 

on behalf of

Dr. Theodore Raymond Muth 

Academic Editor

PLOS ONE